# Determinants of Vaccination Uptake in Risk Populations: A Comprehensive Literature Review

**DOI:** 10.3390/vaccines8030480

**Published:** 2020-08-27

**Authors:** Laura Doornekamp, Leanne van Leeuwen, Eric van Gorp, Helene Voeten, Marco Goeijenbier

**Affiliations:** 1Department of Viroscience, Erasmus MC, University Medical Center Rotterdam, Doctor Molewaterplein 40, 3015 GD Rotterdam, The Netherlands; l.p.m.vanleeuwen@erasmusmc.nl (L.v.L.); e.vangorp@erasmusmc.nl (E.v.G.); 2Travel Clinic, Erasmus MC, University Medical Center Rotterdam, Zimmermanweg 7, 3015 CP Rotterdam, The Netherlands; 3Department of Infectious Diseases, Erasmus MC, University Medical Center Rotterdam, Doctor Molewaterplein 40, 3015 GD Rotterdam, The Netherlands; 4Municipal Public Health Service Rotterdam-Rijnmond, Schiedamsedijk 95, 3011 EN Rotterdam, The Netherlands; hacm.voeten@rotterdam.nl; 5Department of Public Health, Erasmus MC, University Medical Center Rotterdam, Doctor Molewaterplein 40, 3015 GD Rotterdam, The Netherlands; 6Department of Internal Medicine, Erasmus MC, University Medical Center Rotterdam, Doctor Molewaterplein 40, 3015 GD Rotterdam, The Netherlands; m.goeijenbier@erasmusmc.nl

**Keywords:** vaccination uptake, vaccine refusal, vaccine hesitancy, risk groups, immunocompromised, travellers, healthcare workers, health behaviour model, determinants

## Abstract

Vaccination uptake has decreased globally in recent years, with a subsequent rise of vaccine-preventable diseases. Travellers, immunocompromised patients (ICP), and healthcare workers (HCW) are groups at increased risk for (severe) infectious diseases due to their behaviour, health, or occupation, respectively. While targeted vaccination guidelines are available, vaccination uptake seems low. In this review, we give a comprehensive overview of determinants—based on the integrated change model—predicting vaccination uptake in these groups. In travellers, low perceived risk of infection and low awareness of vaccination recommendations contributed to low uptake. Additionally, ICP were often unaware of the recommended vaccinations. A physician’s recommendation is strongly correlated with higher uptake. Furthermore, ICP appeared to be mainly concerned about the risks of vaccination and fear of deterioration of their underlying disease. For HCW, perceived risk of (the severity of) infection for themselves and for their patients together with perceived benefits of vaccination contribute most to their vaccination behaviour. As the determinants that affect uptake are numerous and diverse, we argue that future studies and interventions should be based on multifactorial health behaviour models, especially for travellers and ICP as only a limited number of such studies is available yet.

## 1. Introduction

Vaccinations have proven to play a major role in the prevention and control of many infectious diseases. However, in the twenty-first century, vaccination programs face multiple challenges [1]. The first one is the need for fast development of effective and safe vaccines for new (re-)emerging pathogens. The recent SARS-CoV-2 pandemic is an example in which a vaccination is highly desired and may reduce the enormous impact of the current pandemic. The second challenge in the field of vaccinology is the upcoming trend of vaccine hesitancy and declining vaccination uptake.

Vaccine hesitancy is recognised by the World Health Organization (WHO) to be one of the ten threats to global health [2]. Vaccination uptake is declining globally, resulting in a rise in outbreaks of vaccine-preventable diseases (VPD) [3]. For instance, measles cases have increased—up to 300 percent—over the past years [4]. Vaccine hesitancy has predominantly received attention in the light of parents rejecting the national immunization programs. However, low vaccination uptake among adult populations also raises concerns [5]. Adults are progressively at risk for infectious diseases because life expectancy increases [6], the incidence of chronic diseases that require immunosuppressive treatment rises [7], and international travel expands [8]. Other determinants will play a role in vaccination uptake in adult populations as compared to children.

Adults who are recommended to get vaccinated can be divided into several risk groups. Risk populations in this context are defined as groups of human individuals with an increased risk of acquiring a (severe) infection due to their behaviour, health, or occupation. To get a broad overview of determinants that play a role in the vaccination uptake among risk groups, this review will focus on three distinct risk groups which consult vaccination clinics frequently, namely: “travellers, immunocompromised patients (ICP) and healthcare workers (HCW)”.

Travellers comprise a risk population, as at their destinations they can be exposed to infectious diseases they have not encountered before. Traveller vaccination guidelines are available to protect this population. These guidelines do not only differ per destination but are also dependent on the activities the travellers will undertake and the duration of their stay. Additionally, the country of origin is of importance, because of the endemicity of infectious diseases and therefore natural exposure, and national immunization programs. Moreover, travellers who are not properly vaccinated for their trip are not only at risk for getting sick themselves, they can also create a public health concern for communicable diseases, as they could carry an infection back home to a naïve population [9].

ICP have an increased risk for serious illnesses caused by infectious diseases due to a diminished function of their immune system. The compromised state of their immune system can be induced by either an underlying disease or the treatment of a disease. As a consequence of fast-developing immunosuppressive therapies for e.g., auto-immune diseases and malignancies, ICP are a constantly growing population [7]. Therefore, optimal protection of this vulnerable group is of utmost importance.

HCW are another risk category for acquiring infectious diseases. Their occupation brings them in close contact with patients, that possibly carry an infectious disease. Furthermore, HCW are not only personally at risk, they may also put their—mostly vulnerable—patients at risk when they work while carrying an infection [10]. On top of that, HCW play an important role in providing their (immunocompromised or travelling) patients with information or recommendations regarding vaccinations.

Vaccination uptake varies between risk populations and there may be differences in determinants that play a role in this behaviour. To find general patterns each risk group will be studied separately. However, as travellers, ICP, and HCW are interrelated, we aim to learn from similarities and differences between these groups. If we understand risk populations’ motivations and concerns, we might be able to address these either separately or combined by effective interventions. To get a better overview of all determinants that have a possible impact on uptake, we classified these in a model of health behaviour change.

An abundance of behaviour change models are available that describe determinants affecting preventive health behaviour [11]. In 2003, the integrated change (I-Change) model was developed by de Vries et al. [12]. This model is derived from the attitude-social norm-self-efficacy (ASE) model and integrates several other models, among which are the often-used health belief model (HBM) and the theory of planned behaviour (TPB) (Appendix A). According to the I-Change model, vaccination behaviour is shaped by the intention to get vaccinated which is subject to barriers and facilitators. Intention is established by motivation, awareness, information, and predisposing determinants. As this I-Change model comprises a wide variety of determinants that are used by other studies, for example those based on the HBM and ASE model, we use this model as a conceptual framework.

With this comprehensive review, we aim to better understand determinants that play a role in the uptake of vaccinations in travellers, ICP, and HCW and explore similarities and differences in these three groups. Hereby, we aim to create a solid ground for the development of evidence-based interventions to increase vaccination uptake in the populations that need optimal prevention strategies for infectious diseases.

## 2. Methods

### 2.1. Search Strategy

We performed a systematic database search on 19 February 2020. We performed one search for all three risk groups (Appendix A). For each risk groups we combined search terms for vaccination uptake and health behavioural models. We searched the following databases: Embase, Medline, Cinahl, Web of Science Core Collection, ERIC, PsychINFO, and SocINDEX. As determinants of vaccination uptake may vary over time, we limited our search to studies published during the last ten years (between 1 January 2010 and 1 January 2020). We excluded research papers written in another language than English. All records were retrieved into an EndNote database. Duplicates were removed and titles and abstracts were screened (by LD). Thereafter, papers were sorted in the three different groups and full texts articles were reviewed for suitability using inclusion and exclusion criteria (by L.D. and L.v.L.) using EndNote X9.

### 2.2. Study Selection

Studies were included if they met all of the following criteria: (1) at least 75% of the included respondents are either ICP (patients with autoimmune diseases, malignancies, HIV, asplenia and solid organ or stem cell transplantations) or travellers (including travellers visiting friends and relatives (VFR), short- and long-term business travellers) or HCW (including general practitioners (GPs), physicians and nurses working in a hospital); (2) addressing self-reported cognitive determinants that may explain vaccination uptake; and (3) being performed in Western countries (defined as Europe, North America, Australia, and New Zealand).

We excluded studies that focussed on: (1) children; (2) HCW who care for populations other than the ICP defined in our study (e.g., paediatricians, elderly home physicians) or who are not directly involved in the care for this group (e.g., pharmacists, dentists); (3) future healthcare workers (e.g., medicine or nursing students); (4) uptake of the national immunization programme (e.g., HPV vaccination); (5) hypothetical vaccinations (e.g., a HIV vaccine); (6) vaccinations administered in outbreak situations (e.g., H1N1 vaccine, Ebola vaccine); (7) other very specific target groups (e.g., Roma travellers, migrants, pregnant women; and (8) predisposing factors exclusively. We also excluded qualitative studies and non-peer reviewed articles such as conference abstracts.

In case any doubt or disagreement between the two researchers who performed the study selection (by L.D. and L.v.L.) arose, the specific papers were discussed in a plenary session with all co-authors.

### 2.3. Data Extraction

The following background characteristics from included studies were extracted: first author and year of publication; study design; enrolment period; enrolment site; sample size; study population; theoretical framework; and targeted outcome variables. Extracted data was collected in Microsoft Excel 2016 and the presence and impact of determinants were rated in separate sheets per study group (by L.D. and L.v.L.). Random samples were taken to check the data extraction and disagreements were discussed plenary with all co-authors. Furthermore, the quality of studies was assessed using the the AXIS tool [13], which is a screening tool specifically designed for cross-sectional studies, as those in our review, and includes 20 items relevant to this design. Scores 1–9 are rates as low, 10–14 as medium and 15–20 as high.

### 2.4. Labelling of Determinants

The I-Change model was used to organize all determinants that could explain vaccination uptake. A simplified version of this model is shown in Figure 1. The following concepts are used: (1) predisposing factors, including baseline characteristics of studied populations; (2) information factors, including information retrieved via media, social contacts and HCW; (3) awareness, of the infectious agent being present or a vaccine being available; (4) knowledge (either examined or self-evaluated), about the consequences of the infection, or about the efficacy and duration of protection of vaccination; (5a) perceived risk of the infection, which is divided into perceived severity of the disease and perceived susceptibility to get infected; (5b) perceived risk of vaccination, including vaccine-specific considerations such as fear of side-effects and trust in the effectiveness of the vaccine; (6) attitude, defined as a person’s disposition to respond favourably or unfavourably to vaccinations [14], often reflected by a person’s general believes about vaccinations; (7) social influence, which can be social norms imposed by family, friends or religion, but also recommendations from a healthcare professional or tour guide; (8) self-efficacy, defined as beliefs in one’s own capacity to perform certain behaviour [15]; (9) intention to behaviour, expressed by people before they perform the behaviour; (10) barriers and facilitators, that withhold individuals from or enable them to certain behaviour, such as time, costs, or accessibility.

## 3. Results

The literature search generated 2227 hits (Figure 2). After removing duplicates and excluding articles published before 2010, 1260 articles were available on the topic. These were screened based on title and abstract, resulting in 242 articles that were eligible for full-text assessment. These were divided into the three subgroups (some were included in more than one category): 30 for travellers, 95 for ICP, and 122 for HCW. Finally, 17, 29, and 44 articles were included in the data analysis for the three groups, respectively. The most common reason for exclusion was that no determinants (other than predisposing factors) were reported. Table 1 describes the characteristics and quality of included studies for travellers, ICP, and HCW. Determinants that play a role in vaccination uptake were retrieved from the articles and summarized in Table 2, Table 3 and Table 4 for travellers, ICP, and HCW respectively. The results of the quality assessment are presented in Appendix A.

### 3.1. Vaccination Uptake Among Travellers

The 17 articles that studied determinants of vaccination uptake among travellers comprised 12 cross-sectional surveys, two pre- and post-travel surveys, and three retrospective studies of which one was based on confirmed cases of VPD (Table 1). Travellers that were studied originated from the USA (6 studies), Australia (4 studies), Europe (5 studies), or mixed continents (2 studies). Sample sizes ranged from 55 to 27,386 and comprised Hajj pilgrims in three studies, travellers to Africa in two studies and to Asia in two studies. Other studies had broader inclusion criteria. Three studies used KAP (knowledge-attitude-practices) surveys and one study mentioned a health behavioural model (theory of planned behaviour) as theoretical background for their study.

#### 3.1.1. Predisposing Factors

Ten articles studied baseline characteristics of travellers that could be associated with vaccination uptake (Table 2). The vaccinations that were studied were diverse, most papers discussed vaccinations for influenza (*n* = 7), hepatitis B virus (HBV) (*n* = 6), hepatitis A virus (HAV) (*n* = 5) and meningococcal disease (*n* = 5). Regarding age, three papers reported that younger people had a higher uptake [18,20,24]. However, for influenza vaccination this was the opposite: older travellers were more likely to be vaccinated for seasonal influenza [27,32]. Gender was not a significant predictor of vaccination uptake in any of the studies. Education level was studied by three papers [18,27,31]. Two found this determinant to be positively associated with (intention to) obtaining recommended vaccinations [28,31]. Seven studies reported travel purpose in relation to vaccination uptake, but the results were diverse. One study concluded vaccination uptake was highest if the reason of travelling was business or backpacking [20]. However, work-related travel was associated with lower uptake in another study (OR = 0.39, (0.17–0.92)) [27]. Travellers visiting friends and relatives (VFR) had a lower uptake in two studies [24,29], but two other studies found no association [20,25]. Six papers studied the relation between travel duration and vaccination uptake. Two studies showed that uptake was significantly lower when people travelled longer [24,28], while one found that it was higher (for rabies only) [29] and three studies found no difference [19,20,27].

#### 3.1.2. Information Factors

No clear relationship between information sources and vaccination uptake was reported. However, eight studies reported a role for the GP, of which three said that the GP was very influential [22,29,30,32].

#### 3.1.3. Cognitive Determinants

Of all the cognitive determinants studied, perceived risk of infection was most frequently described in relation to vaccination uptake (*n* = 10). Only one study found a significant positive relation (OR 1.74 (95% CI 1.14–2.62)) [16], and another five reported this factor to play a role in the majority of the study population. Although not often tested for significance, “not feeling at risk of the disease” was a common explanation of a lot of travellers for not receiving the recommended vaccinations. Perceived risk of vaccination was sparsely discussed (*n* = 4).

Social influence, which comprises mostly trust and recommendations of healthcare providers in this selection of studies, was reported in seven papers and was recognised as important by the majority of the study population in four papers.

Attitude was described in six papers, and was not found to be significant in two of them [19,31]; reliance on natural immunity was mentioned three times as a reason to reject vaccination [17,23,30]. Awareness was also discussed in six papers; although it was not tested for significance, 13–73% mentioned unawareness of the availability of the vaccination (or unawareness of the recommendation of the vaccination) as an important reason for non-uptake [17,18,20,21,22,30].

Five studies reported on knowledge of VPD; two found a significant positive relation between knowledge and vaccination uptake [20,26], one found no relation [19].

#### 3.1.4. Barriers and Facilitators

Reported barriers could be classified in costs and lack of time. Costs were the most described; however, it played a modest role in explaining non-uptake and differed per vaccination. For instance, for influenza vaccination uptake costs were mentioned to play a role in less than 7% of travellers, while for HBV (12%), Japanese encephalitis (35%) and pneumococcal vaccination (38%) concerns about costs were much higher. In two papers lack of time was given as part of the explanation of non-uptake in more than 10% of the study population [17,22]. One paper described that 3–24% of travellers require a reminder to complete their vaccination series [22].

### 3.2. Vaccination Uptake among Immunocompromised Patients

Twenty-nine articles concerning ICP were included. Most of these studies were cross-sectional (*n* = 23), but four were prospective (with a follow-up moment) and two retrospective (Table 1). Studies were performed among European (*n* = 23), American (*n* = 3) and Canadian (*n* = 3) populations. Sixteen studies involved patients with auto-immune diseases, of which four studies focussed completely on patients with inflammatory bowel disease. The vaccination uptake of HIV patients was studied in three papers. Four papers studied populations with solid tumours, six papers studied patients who received haematological stem cell transplantation (HSCT) and three papers investigated patients who received a solid organ transplantation (SOT). Almost all papers addressed the influenza vaccination uptake (*n* = 25) and many also included the uptake of pneumococcal vaccinations (*n* = 13). Influenza vaccination rates varied from 6–79% and pneumococcal vaccination rates from 2–54%. Lowest rates were reported in Polish inflammatory bowel disease (IBD) patients [60] and highest in American rheumatic patients [55]. In ICP, health behaviour models were cited slightly more than in the travellers population. Two studies were based on the (HBM) and another three studies used KAP surveys.

#### 3.2.1. Predisposing Factors

Most studies (17 out of 24 that studied age) found a positive association between age and vaccination uptake (Table 3). Especially for influenza vaccination, older patients tend to be more compliant with vaccination guidelines in the studied year. Only in one study a negative association was found (OR 0.02, 95% CI (0.01–0.57)) [46]. Most studies report that gender and education level are not significant predictors of vaccination uptake in ICP, with a few exceptions. Three studies showed in a multivariate analysis that males had a higher uptake. Two studies showed a negative association between uptake and education level, while one showed a positive association. In five studies, the use of strong immunosuppressive medication was positively associated with vaccination uptake, whereas in two studies the association was negative and in three there was no association. Generally, ICP with comorbidities in their medical history tend to have a higher uptake in four [38,39,42,54] out of seven studies. One study reported a negative association [42] and two found no significant difference [33,52]. All five papers that included vaccination history (for the same or another vaccination), concluded that there was a positive association between vaccination uptake in the past and current uptake [34,43,46,47,52].

#### 3.2.2. Information Factors

Thirteen studies investigated where ICP retrieve their information from. In general, gathering information from online media sources was somewhat associated with a lower vaccination uptake, while receiving information from HCW resulted in a higher uptake [35,41].

#### 3.2.3. Cognitive Determinants

Perceived risk of vaccination was the most frequently mentioned cognitive determinant, being discussed in 21 of the 29 articles. In all three papers that tested for significance, a negative correlation with vaccination uptake was found, meaning that a higher perceived risk of a vaccine results in a lower uptake. But also that a lower perceived risk, reflected for example by trust in the effectivity of this specific vaccine, increases the uptake. Fear for side-effects or deterioration of their disease caused by the vaccination were mentioned often. Another concern that was often expressed was the doubt of effectivity of vaccination, due to either the immunogenicity of the vaccine or due to the compromised state of the patients’ immune system. Distrust was reported more often for influenza than for other vaccinations [55].

Awareness of either the availability of or the indication for a vaccination was also widely discussed (*n* = 17). While only found to be significantly correlated twice, this determinant played a role in the majority of the study population in seven papers. Because ICP often mention vaccination not being proposed as a reason for non-uptake, this determinant is related to the information factors, knowledge, and HCW recommendation.

Attitude, covering the attitude to vaccinations in general, was mentioned in 14 studies and was found to be positively correlated twice in multivariate analysis. The effect of a favourable attitude to vaccinations in general was larger on uptake of influenza (adjusted odds ratio (aOR) 3.4 (95% confidence interval (CI) 1.2–9.5)) than on uptake of pneumococcal vaccination (aOR 1.7 [95% CI 0.8–3.5]) [44]. Perceived risk of infection was mentioned equally often as attitude (*n* = 14) and was also positively associated with uptake, in two of the four studies that tested for significance [46,59].

Although knowledge was only addressed in four papers, in two out of the three articles that tested for significance a positive correlation was found. Recommendation of an HCW was studied in 12 out of the 29 papers and a significant correlation was found in all eight papers that performed statistical analysis. In addition, a frequently reported reason for not being vaccinated was that vaccination was not offered or recommended, which we included under awareness.

Self-efficacy was reported in two papers. One reported that more than 10% of unvaccinated ICP were unsure of how to arrange to receive the vaccines [56], while another reported that patients who find it easier to attend a GP for vaccination, have a higher intention to get vaccinated (*p* < 0.001) [46]. Regarding intention to behaviour, one high-quality study expressed that 80% of their IBD study population expressed to be willing to receive all of the recommended vaccinations, while only 9% had ever received a pneumococcal vaccination and only 28% was vaccinated against influenza at the time of participation in the study [58]. In another study with 17% influenza and 4% pneumococcal vaccination uptake, the intention to be vaccinated next year was also high and not significantly different between the vaccinated (89%) and unvaccinated group (80%) [59].

#### 3.2.4. Barriers and Facilitators

Cost was only mentioned as a barrier in one paper that found a significant negative correlation with uptake [36]. Lack of time (*n* = 2) and the inconvenience of another appointment (*n* = 4) were more often given as reasons for declining vaccination.

### 3.3. Vaccination Uptake among Healthcare Workers

In HCW, influenza vaccination uptake is most widely studied. In 35 articles out of the 44, seasonal influenza vaccination was the only vaccine studied, with uptake varying between 9% [63] to 97% (mandatory policy) [96]. Most studies were conducted in Italy (*n* = 8), followed by France (*n* = 5) and the USA (*n* = 5). All but one were designed as cross-sectional surveys, with sample sizes ranging from 77 [76] to 32,808 [91]. Seven studies mentioned the use of a theoretical model for their study, which includes the HBM [88], the TPB [89], the risk perception attitude framework [93], the Triandis model of interpersonal behaviour [81], the cognitive model of empowerment [99] or mixtures of different models [64,79] (Table 1).

#### 3.3.1. Predisposing Factors

Thirty-six articles studied at least one predisposing factor in relation to vaccination uptake (Table 4). Of the 30 articles that studied age, 22 found that older healthcare workers had a significantly higher uptake. On the other hand, in the case of hepatitis B [84,95] and measles [78,82], younger HCW’s had higher compliance. In the 27 papers that studied gender, being male was associated with higher vaccination uptake in 13 studies. Five papers mentioned a significantly higher uptake in women, one for rubella only [97], and another for hepatitis B only [82]. Occupation was studied in relation to vaccination uptake in 18 articles. Sixteen papers showed that physicians had a significantly higher uptake than other HCW. This also complies with the significant positive association between education level and uptake that was found in five papers. Presence of a chronic disease resulted in significantly higher uptake in seven studies. In three other studies investigating this factor, no association was found. Having children at home was studied in nine papers, but six found no significant role for this factor in vaccination uptake. Good vaccine compliance in the past turned out to be an excellent predictor of uptake in all 11 studies investigating this factor.

#### 3.3.2. Information Factors

The role of information sources in vaccination uptake was studied in six articles. When information was gathered from evidence-based sources, uptake was significantly higher in all five studies that investigated this source. On the other hand, uptake was lower when information was retrieved from social media, television, or radio [63,92]. Only one study found that gaining information from colleagues was associated with a higher uptake [78].

#### 3.3.3. Cognitive Determinants

Perceived risk was the most frequently described determinant in HCWs. More specifically, perceived personal risk of infection reflects the perceived risk to contract the VPD, including the perceived susceptibility to get infected and the perceived severity of the disease if contracted. In 33 out of 35 papers mentioning perceived risk of infection, a significant positive relation was found between this determinant and vaccination uptake (*n* = 13), or these reasons were mentioned in a considerable part of the study group (*n* = 20). Furthermore, in 18 papers a high perceived risk to infect patients was given as a reason for vaccination uptake. Perceived risk (vs. benefit) of vaccination was mentioned in 34 papers. Fifteen studies reported a significant negative relation between perceived risk and uptake, indicating that high perceived risk or low perceived benefit of the vaccination resulted in lower uptake. Additionally, five papers mentioned that this determinant played a role in the majority of the study population. Adequate knowledge of recommendations, effectiveness, and side-effects of vaccinations was significantly positively associated with uptake in 11 papers; in four studies, no significant association was found. Attitude towards vaccination was studied in 22 articles. In half of them, a significant positive association with vaccination uptake was found. Social influence (encouragement of colleagues, managers, family) was analysed in almost half of the studies (*n* = 15). In only one study no association was found [66], but the others showed either a significant (*n* = 8) or considerable (*n* = 6) positive relation with vaccination uptake. Specific for HCW are the social arguments ‘I got vaccinated because it’s my duty as an HCW’ or ‘as an HCW, I have a role in the prevention of epidemics/spread of diseases’, that we collected under the term ‘professional norms’. This determinant was positively associated with uptake in all 15 studies focusing on this factor; in seven out of 11 studies that tested for significance, this factor remained a strong predictor for uptake in multivariate analysis.

#### 3.3.4. Barriers and Facilitators

In comparison with the previous determinants, barriers and facilitators are relatively less studied. Of the barriers, time-related factors were mentioned most frequently and played a considerable role (>10%) in hindering uptake in seven studies. Costs turned out to be no barrier. The fact that the vaccines were free of charge even appeared to be a reason for uptake in two studies [62,67]. On the other hand, facilitators stimulating uptake were getting a reminder (*n* = 3), convenient time/place of distribution (*n* = 4), and getting a reward (*n* = 3). However, in none of the studies were the potential rewards specified.

## 4. Discussion and Conclusions

Our review of the currently available literature shows that there are clear differences in determinants that play a role in vaccination uptake in travellers, ICP, and HCW. For travellers, low perceived risk of infection and low awareness of vaccination recommendations are most accountable for low uptake. For ICP, awareness of the indication of vaccination plays an important role, together with receiving vaccination recommendations from their treating physician. ICP have a high perceived risk of vaccination, due to not only fear for general side-effects but also concerns about potential consequences for their illness. For HCW, perceived risk of (the severity of) infection for themselves and for their patients together with perceived benefits of vaccination contribute most to their vaccination behaviour.

Regarding predisposing factors, there is a clear positive relationship between age and influenza vaccination uptake in all risk groups. This could be explained by the additional indication older people have for influenza vaccination. However, for other vaccinations, this relationship is either inverted or non-existent. Higher vaccination uptake was seen in males in HCW and ICP, which could be associated with the fact that females worry more about vaccine safety and efficacy than males [107]. Indeed, more side-effects are reported by females, while on the other hand, from a biological perspective, females typically mount higher antibody responses [107]. Although we did not find a clear relationship between education level and vaccination uptake in the risk groups, in HCW the uptake was markedly higher in physicians compared to other HCW. Overall, vaccination history seems to be an excellent universal predictor of future vaccination uptake, probably due to unaltered cognitive determinants.

Regarding cognitive determinants, the greatest diversity between risk groups was found in awareness. In ICP, almost two-thirds of the studies mentioned limited awareness, compared to one-third in travellers and none in HCW. With their education and occupation, it seems quite obvious that HCW are aware of the opportunities and indications for vaccinations. The fact that ICP seem less aware than travellers might have to do with travellers taking an active decision to go abroad realizing that they have to prepare themselves, while patients get passively diagnosed with a disease, and are more dependant of the HCW for information provision. In all groups, HCW as a source of information has a positive effect on uptake. The strong relationship between HCW recommendations and vaccination uptake in ICP (reaching odds ratios up to 53 [52] and 187 [44]), underline the importance of positive attitudes towards vaccination in HCW themselves [100,108].

In general, knowledge has a positive influence on uptake in all risk groups. However, since several studies showed no relation between knowledge and uptake [19,35,62,66,71,95], improving education alone will probably not be sufficient to increase uptake. In all groups, the perceived susceptibility and severity of diseases on one hand and the perceived effectiveness and risks of vaccinations on the other hand are important determinants predicting uptake. Especially ICP and HCW express concerns about the safety and effectiveness of vaccines particularly for influenza vaccination [38,44]. And although the effectiveness of influenza vaccination varies with the coverage of circulating strains each year, another part of the perceived lack of effectiveness could also be explained by the lack of protection for other common cold viruses that can cause influenza-like symptoms [109]. Travelers seem to have low risk perceptions for the diseases they could be vaccinated for as well as for the potential negative effects of vaccination. Despite the high morbidity and mortality of some VPD such as yellow fever, hepatitis B, and influenza, in all risk groups, some participants stated they preferred natural immunization or were against vaccinations in general. Remarkably, attitudes differ for specific vaccinations, for instance, people tend to have a more positive attitude towards pneumococcal vaccination in comparison to the seasonal influenza vaccination [55]. Interestingly, the mistrust of ICP and HCW towards the vaccinations produced by the pharmaceutical industry seems disproportionate to therapeutics manufactured by the same pharmaceutical companies [40,50,72,78]. Here, the difference between prevention and treatment might play a role, where the latter provides a more direct and visible effect. Another possible reason for the negative general attitude towards vaccination, also described in decision making for childhood vaccinations [110], is the increasing tendency for self-empowerment towards personal health decisions. In this view, individuals stand up against imposed policies and want to make their own decisions, which could also be judged by peers as independent and smart decision making [110,111]. At the same time, sources that are being used to make personal health decisions, such as the internet, contain a lot of negative stories [112].

Practical barriers and facilitators play a limited role in vaccination uptake compared to the other determinants. In all three groups, a reminder is an important facilitator and (lack of) time an important barrier. Especially for HCW, this factor is interesting. Physicians report this factor most frequently [73]. They do not only experience lack of time to get vaccinated, they also feel that lack of time impedes their duty to recommend vaccinations to their patients [113]. Again, as HCW recommendations are strongly positively associated with uptake, not only in the other risk groups, but also for HCW themselves (by colleagues for example) [66,80], removing this barrier can result in achieving optimal care for all groups.

Only 16 of the 90 articles that were analysed in this review were based on a health behaviour model. Many of those found determinants which contributed to vaccination uptake to a greater or lesser extent [46,64,79,93,99]. Interventions that focus on a single determinant, such as knowledge, repeatedly proved to be ineffective in the past [66], while multifactorial cognitive intervention strategies are effective to improve uptake [114,115]. Therefore, all determinants that play a role have to be taken into account. Predisposing factors could be used to target specific subgroups and personalize uptake strategies [93]. Facilitators and barriers could be added or taken away to increase vaccination uptake. But, most importantly, interventions need to address cognitive determinants. Interventions that increase awareness and risk perception of infectious diseases are more effective than those decreasing risk perceptions of vaccination by providing scientific information [116]. Social norms can be influenced in the case of hierarchical relationships, for instance, the employer will have an effect on the vaccination decision of HCW and HCW will impact ICP’s decisions. Therefore, multifactorial interventions are needed that address the most important cognitive determinants. As these include awareness and risk perceptions, reminders and incidence data could help. Reminders for travellers could be disseminated in general media before holidays, while for ICP patient associations and HCW could play a role. To improve risk perceptions for the infections, cases of vaccine-preventable diseases should be made public. To decrease risk perceptions of negative effects of vaccinations (e.g., adverse events) new studies should compare the number of influenza-like illnesses in vaccinated and non-vaccinated groups. Furthermore, social norms can be included by making the decisions of vaccination uptake public. For example, in HCW trials have been implemented to test the effects of providing a pin that vaccinated HCW may wear that is saying “deliberately vaccinated”, which could affect both colleagues and patients [117].

Vaccination decisions of travellers and ICP are less well studied than those of HCW. Additionally, data on uptake of vaccinations other than influenza are limited. As the available data show large differences in determinants predicting uptake of influenza versus other vaccinations, further studies are required regarding the uptake of recommended vaccinations for diseases other than influenza. Reaching a more comprehensive understanding of vaccination uptake in different risk groups for the different vaccinations that are indicated, interventions can be developed based on evidence. Moreover, this understanding could help with the implementation of new vaccines for certain risk groups, for instance when a novel SARS-CoV-2 vaccine will be recommended for HCW.

A number of limitations have to be taken into account when interpreting the results of this review. First, articles were only included if they discussed any cognitive determinants that were possibly related to vaccination uptake. This resulted in the exclusion of papers that looked only, although thoroughly, into predisposing factors. Secondly, there was a high level of heterogeneity in the determinants reported, as studies used various health behaviour models as a framework for their studies, and many did not even use a model but just reported results of questionnaires with either open-ended or multiple-choice questions. Furthermore, the influence of determinants on vaccination uptake was measured with different statistical analyses, which also contributed to the high heterogeneity of the data. Therefore, we choose to report the significance and direction of the association, instead of the magnitude. In addition, we choose to compare three different risk groups that we think are important, thereby we could not discuss all determinants in depth. Finally, included studies were based on self-reported vaccination behaviour. Therefore, we have to take into account a certain level of social desirability and recall bias.

To our knowledge, this is the first review that provides a comprehensive overview of health behavioural determinants explaining vaccination uptake in three different risk groups, namely travellers, ICP, and HCW. We showed that there is a large diversity of determinants that affect uptake to a greater or lesser extent. Therefore, we argue that future studies and interventions should be based on multifactorial health behaviour models, especially for travellers and ICP as only a limited number of such studies is available yet.

## Figures and Tables

**Figure 1 vaccines-08-00480-f001:**
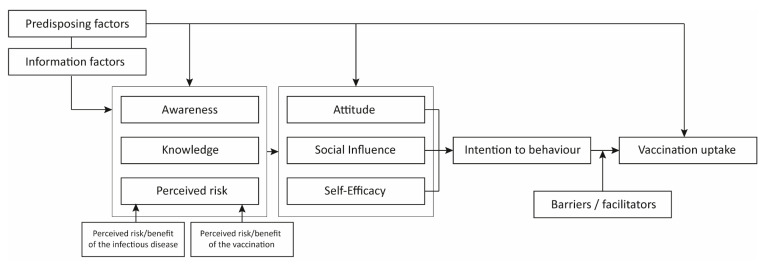
Simplified I-Change model summarizing the studied determinants that could predict vaccination uptake. We used a simplified version of the I-Change model applied to vaccination uptake. Uptake is shaped by the intention to get vaccinated which is subject to barriers and facilitators. Intention is established by motivation (attitude, social influence, and self-efficacy), awareness (awareness, knowledge, and perceived risk) and information and predisposing determinants. Predisposing factors include baseline characteristics of studied populations and influence awareness, motivation and uptake. Information factors include information retrieved via media, social contacts and healthcare workers.

**Figure 2 vaccines-08-00480-f002:**
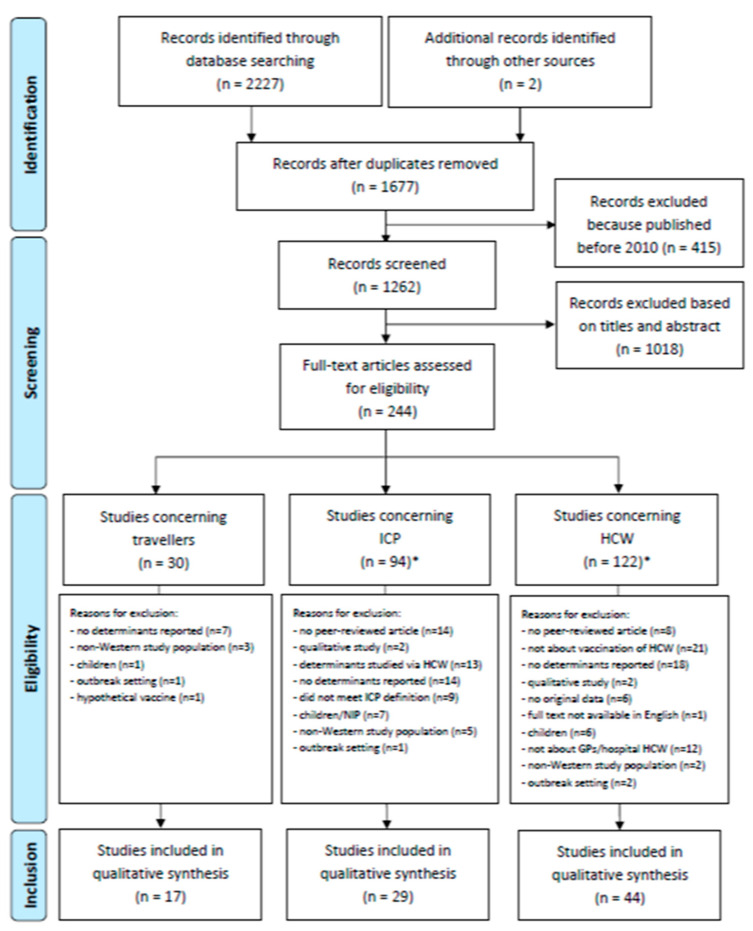
Flow diagram of study selection procedure. * *n* = 2 articles were included in both ICP and HCW.

**Table 1 vaccines-08-00480-t001:** Study characteristics of included studies for travellers, ICP and HCW.

Study	Study Design	Enrolment Period	Enrolment Site	Sample Size	Study Population	Theoretical Framework	Outcome Measures *	Vaccination Coverage	Quality Score **
Balaban, 2013 [16]	Pre- and post-travel surveys	2009	USA	186	American Hajj pilgrims	None	Seasonal influenza	-	Low
Barasheed, 2014 [17]	Cross-sectional survey	2011–2012	Mina (Mecca)	966	Australian Hajj pilgrims	None	Influenza	62%	High
Duffy, 2013 [18]	Cross-sectional survey	2007 (Aug.–Sept.)	United States	1691	American travellers to Asia	None	JE	11%	Medium
Frew, 2017 [19]	Cross-sectional survey	2015 (Feb.–March)	Ferry ports of 2 popular islands, Thailand	1680	Backpackers from Europe, Canada, Australia and New-Zealand (94%)	None (KAP)	HBV	31% completed series	High
Goodman, 2014 [20]	Online cross-sectional survey	2010 (Feb.)	UK	302	Travellers to the meningitis belt of Africa in last 3 years or planned to do so next 6 months	None	MenAWCY	30%	Medium
Herbinger, 2011 [21]	Online cross-sectional survey	2009 (Dec.)	Netherlands, Czech Republic, Spain, Sweden	4203	Travellers to countries of moderate or high prevalence for HBV in the last 5 years	None	HBV	39% in the previous 5 years	Medium
Heywood, 2016 [22]	Online cross-sectional survey	2014 (Aug.–Oct.)	Australia, Finland, Germany, Norway, Sweden, UK, Canada	27,386	Travellers (18–65 years) who travelled to HAV endemic countries in Africa, Asia, South/Central America in the last 3 years	None	HAV/HBV	27% for 3-dose combined HAV and HBV and 37% for 2-dose monovalent HAV schedules	Medium
Igreja, 2019 [23]	Cross-sectional survey	2019 (May–June)	Travel Clinic, Lisbon, Portugal	55	Portuguese travellers	None	Attitudes to vaccinations in general	-	Low
Lammert, 2016 [24]	Retrospective study	2012–2014	clinics from Global TravEpiNet, USA	24,478	International travellers who sought pre-travel health advice	None	Refusal rates of recommended vaccines and reasons	25% refused one or more recommended vaccine(s)	High
Paudel, 2017 [25]	Prospective enhanced surveillance study	2013 (Feb–2014 (Jan.)	Australia	180	confirmed cases of typhoid, paratyphoid, measles, HAV, HEV, chikungunya, malaria	None	Seeking pre-travel advice and uptake	25% sought pre-travel advice and 16% got vaccinated	Medium
Pavli, 2019 [26]	Cross-sectional survey by email	2015 (Nov.)–2016 (Mar.)	Greece	231	Greek (non-healthcare) students from 36 universities, planning to study abroad	None	Men, intention to vaccinate	23% vaccinated, 15% intention	Medium
Pfeil, 2010 [27]	2 cross-sectional surveys	2009 (Jan.–Feb.), 2010 (Jan.)	Centre for Travel Health, Zurich, Switzerland	623	Travellers to a resource-limited destination	None (KAP)	Seasonal (and pandemic) influenza	14% seasonal influenza	High
Selcuk, 2016 [28]	Cross-sectional survey	2013 (July)	Istanbul Ataturk Airport, Istanbul, Turkey	124	Turkish travellers to Africa	None	Recommended for destination	53% vaccinated pre-travel	Medium
Tan, 2017 [29]	Retrospective cohort study	2012 (Jan.)–2013 (Dec.)	Mayo Clinic Travel and Tropical Medicine Clinic, Minnesota, USA	2073	Children and adults who sought pre-travel advice (19% VFR)	None	Documented receipt or positive serology or completion of series	94% men in VFR, 12% rabies in VFR	High
Tashani, 2016 [30]	Cross-sectional survey	2014–2015	Immunization clinic, Sydney, Australia	300	Travellers (>18 year) planning to attend Hajj	None	Pneu and DTP when recommended	17% pneu, 14% DTP	Medium
Wiemken, 2015 [31]	Cross-sectional study	2013 (Nov.)–2014 (July)	University of Louisville, Travel Clinic, USA	183	American travellers before their consultation	TPB	Intention to get vaccinated	Not given	High
Yanni, 2010 [32]	Pre- and post-travel surveys	2008 (June–Sept.)	Departure lounges at airports in New York, Chicago, Los Angeles, and San Francisco	1301 (pre) (337 post)	American travellers who will travel to Asia	KAP	Influenza	41%	High
Akin, 2016 [33]	Cross-sectional survey	2015 (July–Sep.)	Daycare chemotherapy unit of Hacettepe University Cancer Institute, Ankara, Turkey	229	Adult patients with cancer receiving chemotherapy	None	Adult vaccination coverage (influenza, tetanus, hepatitis, pneu)	54% were vaccinated at least once, only 9% after cancer diagnosis	Medium
Althoff, 2010 [34]	Nested influenza study (interview administered surveys)	2006–2007 and 2007–2008	5 cities in the USA	1462	HIV+ women	HBM	Influenza	55–57% of women reported vaccination (about 44% not vaccinated)	Medium
Battistella, 2019 [35]	Cross-sectional observational study	2017 (Jan.–July)	7 large dialysis services, Italy	703	Dialysis patients	None	Influenza	58% adherence	High
Chehab, 2018 [36]	Cross-sectional study (in longitudinal cohort)	2012 (Nov.)–2013 (Oct.)	Germany	579	SLE patients (48% on IS)	None	Influenza, tetanus, pneu, men and previous refusal	45% influenza (last year); 65% tetanus; 32% pneu; 6% men	High
Chin-Yee, 2011 [37]	Cohort study (one time follow-up)	2009 (Oct.)–2010 (Mar.)	Tertiary care cancer center, Canada	129	Patients with hematologic malignancies (92% chemotherapy, 76% in past 3 mo)	None	Seasonal influenza (and pandemic)	57% seasonal influenza	Medium
Gagneux-Brunon, 2019 [38]	Cross-sectional survey	Unknown	France	468	HIV+ patients	None	Pneu, HAV, HBV, seasonal influenza	30% IPD; 24% HAV; 64% HBV; 40% influenza	Low
Haroon, 2011 [39]	Cross-sectional survey (audit)	2009 (Sept.)	Outpatient clinics, tertiary university hospital, Ireland	110	Rheumatology patients on IS	None	Seasonal influenza and pneu	34% influenza; 11% pneu; 11% both	Medium
Harrison, 2017 [40]	Cross-sectional survey	2015 (Aug.–June)	HIV out-patient department of the University Hospital of Vienna, Austria	455	HIV patients	None	Seasonal influenza	12% influenza	Medium
Harrison, 2018 [41]	Cross-sectional survey	2017 (July–Oct.)	Outpatient clinic, Medical University of Vienna, Austria	490	Inflammatory rheumatic disease patients on IS	None	Seasonal influenza	25% influenza	Medium
Lachenal, 2010 [42]	Cross-sectional survey (standardized questionnaire)	2008 (Jan.)	Centre Léon-Bérard, Lyon, France	200	Patients with haematological malignancies (hospitalized or at outpatient clinic)	None	Influenza	26%	Medium
Loubet, 2015 [43]	Self-reported cross-sectional survey	2013 (Summer)	AVNIR, a group of associations whose goal is to support ICP, France	3653	79% autoimmune, 13% SOT, 8% treated for hematological malignancies. 85% on IS.	KAP	Influenza and pneu	59% seasonal influenza and 49% pneu	Medium
Loubet, 2018 [44]	Self-reported cross-sectional survey	2015 (Dec.)–2016 (March)	AFA, national association of patients with IBD, France	199	IBD patients (62% receiving IS)	KAP	Influenza and pneu	34% influenza, 38% pneu	Medium
Malhi, 2015 [45]	Cross-sectional survey (self-reported, paper-based)	2013 (Sept)–2014 (Jan.)	IBD Clinic or Endoscopy Suite at Mount Sinai Hospital, Toronto, Canada	305	IBD patients (53% using biologicals/steroids)	None	Influenza, pneu, HAV, HBV, VZV, men, HVZ, HPV	61% influenza, 10% pneu, 61% HBV, 52% HAV, 26% VZV, 21% men, 5% HZV, 11% HPV	High
Miller, 2018 [46]	Cross-sectional survey	2016 (June–Sept.)	3 tertiary autologous and allogeneic HSCT centres, UK	93	HSCT patients (79% autologous)	adjusted HBM	Intention to receive seasonal influenza	76% expressed high intent	High
Mouthon, 2010 [47]	Cross-sectional survey (standardized questionnaires)	2006 and 2007	Dept. Of Internal Medicine, Cochin Hospital, France	177	Patients with systemic sclerosis	None	Influenza	39% (last year)	Medium
Narula, 2012 [48]	Cross-sectional survey	2010 (May–Aug.)	McMaster University Medical Centre Digestive Disease Clinic, Canada	250	IBD patients (63% on IS)	None	Seasonal (and H1N1) influenza	25% seasonal influenza	High
Nguyen, 2017 [49]	Cross-sectional survey with invitation RCT for new pneu vaccine)	2014 (Oct.–Nov.)	Outpatients clinic of rheumatology at 2 hospitals in Graasten, Denmark	192	RA patients	None	Influenza and pneu	59% seasonal influenza ever, 49% last year, 6% pneu	High
Poeppl, 2015 [50]	Cross-sectional survey	2013 (July)–2013 (Oct.)	Outpatient departments of the General Hospital Vienna, Austria	444	Patients with malignancies (55% solid tumours, 22% haematological malignancy, and 17% had no diagnosed malignancy)	None	Influenza	18% influenza last year	Medium
Price, 2019 [51]	Cross-sectional survey	2014 (June–July)	Cancer center providing ambulatory care, USA	703	Patients (83%) (and caregivers and family (17%) of patients) treated for malignancies	None	Influenza	Patients 72%, caregivers 71% (last year)	Medium
Restivo, 2017 [52]	Prospective observational study	2014 (Oct.)–2015 (April)	SOT Reference Center in Palermo, Sicilia, Italy	82	SOT recipients during hospital admission for transplantation	None	Influenza	38%	Medium
Ruiz-Cuesta, 2016 [53]	Prospective observational study	2012 (Jan.–March)	Reina Sofía University Hospital, Córdoba, Spain	153	IBD (50% UC, 50% CD) patients (>14 years old), 34% on biologicals/corticosteroids	None	HAV, HBV, VZV, MMR assessed by registry	84%	Medium
Sadlier, 2015 [54]	Retrospective study, with provider-delivered survey	2014 (Jan.–Feb.)	Tertiary university hospital in Ireland	170	Dermatology patients prescribed systemic IS	None	Influenza and pneu	38% seasonal influenza last year, 21% pneu last 5 years, 18% both.	Medium
Sandler, 2016 [55]	Cross-sectional, telephone survey	2013 (July–Sept.)	Memorial Medical Center in Chicago, USA	102	RA patients (85–91% taking IS)	None	Self-reported and EHR influenza, pneu, and HZV	79% influenza last season, 54% pneu and 8% HZV	High
Savage, 2011 [56]	Retrospective audit	2010 (Aug.–Oct.)	Outpatient dermatology clinics in Aberdeen RoyalInfirmary, Scotland	87	Immunocompromised dermatology patients	None	Influenza and pneu	70% influenza (last year), 22% pneu	Medium
Struijk, 2015 [57]	Cross-sectional survey	Unknown	Renal Transplant Unit, Academic Medical Center, Amsterdam, NL	526	77% renal transplant recipients (and their nephrologists)	KAB	Influenza, tetanus, pneumococci, HAV, HBV	56% influenza, 15–30% tetanus, 0–5% pneu, 5–30% HAV, 10–20% HBV	High
Teich, 2011 [58]	Cross-sectional survey	2009 (April–Sept.)	Germany	203	IBD patients who had not received vaccination counseling ≥1 year (54% on IS)	None	Vaccinations in general	67% tetanus (<10 years), 21% pertussis, 28% seasonal influenza, 9% pneu	High
Urun, 2013 [59]	Cross-sectional survey (with face-to-face interviews)	2012 (Jan.–March)	Medical Oncology Department of Ankara University Faculty of Medicine, Turkey	359	Patients with malignancies	None	Influenza and pneu	17% influenza 4% pneumococcal	Medium
Waszczuk, 2018 [60]	Cross-sectional survey (self-completed)	Unknown	Wrowclaw, Poland	195	IBD patients (70% on IS)	None	Influenza, HBV and pneu	HBV 55%; Tdap 12%; HAV 7%; annual influenza 6%; VZV/HZV 3%, and pneu 2%	High
Wilckens, 2011 [61]	Cross-sectional survey	2009 (April–Oct.)	IBD outpatients’ clinic, a tertiary referral center, Lueneburg, Germany	102	IBD patients (57% CD, 91% on IS)	None	Vaccinations in general	19% influenza, 3% pneumoccous, 22% HBV, 5% VZV, 55% MMR, and 63% tetanus. Of those who had traveled, 9% HAV and 1% YF	High
Akan, 2016 [62]	Cross-sectional study	2014 (June–Sept.)	family health care centres in Turkey	596	GPs	used, name not mentioned	Seasonal influenza	27%	High
Asma, 2016 [63]	Cross-sectional study	2015 (Jan.)	6 university hospitals in Turkey	642	177 (28%) physicians and 448 (71%) nurses	None	Seasonal influenza	9%	Medium
Boey, 2018 [64]	Cross-sectional study	2015 (Nov.–Dec.)	13 hospitals and 14 nursing homes in Belgium	5141	4506 hospital staff, 635 HCW nursing home staff.	HBM, HIM and ASE	Seasonal influenza	2014: 62% (hospital) 2015: 65% (hospital)	High
Bonaccorsi, 2015 [65]	Cross-sectional study	2010 (Oct.–Nov.)	Careggi University Teaching Hospital, Florance, Italy	2576	10% physicians, 39% nurses, 23% students, 4% health care assistant, 15% other	None	Seasonal influenza	18%	Medium
Castilla, 2013 [66]	Cross-sectional study	2012 (Mar.–May)	PHC workers, Spain	1956	47% GP, 10% paediatricians, 43% nurses	None	Seasonal influenza	52–61% (2008–2011)	High
Ciftci, 2018 [67]	Cross-sectional study	2015 (Sept.–Dec.)	University Hospital, Ankara, Turkey	470	Tertiary healthcare setting (18% physicians, 29% nurses, 11% assistants, 23% auxillary, 9% paramedics, 10% secretaries)	None	Seasonal influenza	27%	High
Costantino, 2019 [68]	Cross sectional study	Influenza seasons 2016–2019	University Hospital of Palermo, Italy	1237	Hospital HCW that had not received influenza vaccination	None	Seasonal influenza	0%	High
Dedoukou, 2010 [69]	Cross-sectional study	2018 (Oct.–Nov.)	76 PHCs in Greece	1617	PHC: 35% physicians, 32% nurses, 23% paramedical/technical, 8% administrative	None	Seasonal influenza	41%	Medium
deSante, 2010 [70]	Cross-sectional study	2009 (Apr.)	2 tertiary care hospitals in Pennsylvania, USA	227	House officers and attending physicians in emergency/internal medicine depts.	None	Seasonal influenza	94%	Medium
Dominguez, 2013 [71]	Cross-sectional study	2012 (Mar.–May)	PHC workers in 7 Spanish regions	1749	Familiy physician (47%), paediatrician (10%), nurses (43%).	None	Seasonal influenza	51%	High
Durando, 2016 [72]	Cross-sectional study	2013 (Oct.)–2014 (Apr.)	San Martino Teaching Hospital/Scientific Research Institute, Italy	830	HCW	None	Seasonal influenza	26%	High
Ehrenstein, 2010 [73]	Cross-sectional study	2006 (Feb.)	Tertiary care university hospital in Germany	652	HCW (physicians 36%, nurses 42%, administrators 22%)	None	Seasonal influenza	34%	Medium
Giese, 2016 [74]	Cross-sectional study	2013	Ireland	164	HCW in a study group of Irish residents	None	Seasonal influenza	28%	Medium
Gramegna, 2018 [75]	Cross-sectional study	2016	Italy	144	Italian Respiratory Society members		Seasonal influenza	55%	Medium
Gutknecht, 2016 [76]	Cross-sectional study	2016 (Feb.–Mar.)	Poland	77	Physicians	None	Seasonal influenza	-	Low
Hagemeister, 2018 [77]	Cross-sectional study	2015 (June-July)	University Hospital Würzburg, Germany	677	Physicians and nursing staff	None	Seasonal influenza	55%	Medium
Harrison, 2016 [78]	Cross-sectional study	-	Vienna General Hospital, Austria	116	Nursing staff	None	HAV/HBV, DTP/Tdap, MMR, influenza, VZV, men, pneu	Seasonal influenza: 42%; Measles: 60%	Medium
Hopman, 2010 [79]	Cross-sectional study	2008 (Nov.–Dec.)	All 8 University Medical Centers in NL	1238	HCW at medium and high risk for influenza	HBM, BIM, ASE	Seasonal influenza	38%	Medium
Hulo, 2017 [80]	Cross-sectional study	2014	University Hospital Lille, France	344	HCW in the emergency departments and the IC units	None	Seasonal influenza	18%	Medium
Johansen, 2012 [81]	Cross-sectional study	2007 (May)	North and South Dakota	155	Randomly selected nurses (52% hospital, 13% clinic, 12% long term)	Triandis	Seasonal influenza	-	Medium
Kalemaki, 2020 [82]	Cross-sectional study	-	Crete, Greece	260	GPs	None	Seasonal influenza, measlesHBV, Tdap	Seasonal influenza 57%; Measles 26% HBV 68%; Tdap 47%	High
Karlsson, 2019 [83]	Cross-sectional study	-	Public hospitals in Finland	2962	Hospital personnel who may work with vaccinations (14% physicians)	None	Seasonal influenza	-	High
Kisic-Tepavcevic, 2017 [84]	Cross-sectional study	2015 (Dec.)	Clinical Centre of Serbia, Belgrade, Serbia	352	HCW	None	HBV	66%	High
Lehmann, 2015 [85]	Cross-sectional study	2013 (Feb.–Apr.)	20 hospitals in Belgium, Germany and NL	1022	56% nurse, 15% physicians, 14% paramedics	None	Seasonal influenza	Total: 37%; Netherlands: 28%; Belgium: 53%; Germany: 36%	High
Maridor, 2017 [86]	Cross-sectional study	2013	3 medium-sized, non-teaching hospitals, Switzerland	252	Nursing staff	None	Seasonal influenza	58%	Medium
Napolitano, 2019 [87]	Cross-sectional study	2018 (Sept.–Nov.)	8 hospitals in Italy	531	Random sample of HCWs (29% physicians, 59% nurses)	None	HBV, influenza, MMR, VZV, pertussis	HBV: 98%; DTP: 91%; MMR: 64%; VZV: 59%; TBC: 50%; Influenza: 30%; Men C: 41%	High
Nowrouzi, 2014 [88]	Cross-sectional study	2011 (Sept.–Nov.)	University of Toronto	963	Medical trainee’s (post graduate)	HBM	Seasonal (and pandemic) influenza	Seasonal influenza 69–76% (2008–2010)	High
Pielak, 2010 [89]	Cross-sectional study	2005 (Apr.)	British Columbia, Canada	719	Immunization nurses of all health units and all physicians that administer vaccinations	TPB	Seasonal influenza	-	High
Prematunge, 2014 [90]	Cross-sectional study	2010 (June)	Tertiary care hospital Ontario, Canada	3275	35% nurse, 5% physician, 11% allied HCW’s, 22% administrative/clerical	None	Seasonal (and pandemic) influenza	Seasonal influenza: 74%	Medium
Quan, 2012 [91]	Retrospective cohort study	2006–2011	University of California Irvine Healthcare	32,808	all HCWs	None	Seasonal influenza	44–92% (2007–2011)	Medium
Rabensteiner, 2018 [92]	Cross-sectional study	2016 (Oct.–Dec.)	South Tyearolean Health Service, Italy	4091	13% physicians, 20% administrative, 67% sanitary or executive non-medical staff	None	Seasonal influenza	10%	High
Real, 2013 [93]	Cross-sectional study	-	Academic medical center in Lexington, USA	318	80% clinical, 20% non-clinical	RPA	Seasonal influenza	66% already received the vaccination or planned to get one soon	Medium
Rebmann, 2012 [94]	Cross-sectional study	2011 (Apr.–June)	Saint Louis region, USA	3188	54% non-hospital HCW, 46 % hospital HCW	None	Seasonal (and pandemic) influenza	2010/11: 79%	High
Scatigna, 2017 [95]	Cross-sectional study	2015 (Apr.–May)	San Salvatore Hospital, L’Aquila, Italy	334	Nurses 53%, physicians 23%, other 24%	None	HBV, influenza, MMR, VZV	-	Medium
Surtees, 2018 [96]	Cross-sectional study	2016	Tertiary referral hospital in Victoria, Australia	1835	HCW	None	Seasonal influenza	97%	High
Taddei, 2014 [97]	Cross-sectional study	2011 (June–Oct.)	6 public hospitals in Florence, Italy	436	59% nurses, 21% physicians, 13% nursing assistants, and 7% were midwives	None	MMR, VZV, Pertussis	11% measles, 7% mumps, 17% rubella, 2% VZV, 7% pertussis	Medium
Tanguy, 2011 [98]	Cross-sectional study	2009 (Nov.)–2010 (Feb.)	Tertiary care centre in Pays de la Loire Region, France	532	24% medical staff, 65% nursing staff, 11% ancillary staff	None	Seasonal (and pandemic) influenza	22%	Medium
Vallée-Tourange, 2018 [99]	Cross-sectional study	2014 (June–July)	A single metropolitan hospital group, UK	784	11% physicians, 36% nurses, 30% allied health professionals, 17% assistants	CME	Seasonal influenza	-	Medium
Verger, 2016 [100]	Cross-sectional study	2014 (Apr.–July)	France	1582	GPs	None	Seasonal influenza, DTP, HBV	72% influenza, 84% DTP, 86% HBV	High
Virseda, 2010 [101]	Cross-sectional study	2009 (Dec.)–2010 (Jan.)	University Hospital 12 de Octubre, Madrid, Spain	527	HCW (23% physician, 29% nurse, 19% nursing assistant, 29% ancillary staff)	None	Seasonal (and pandemic) influenza	50%	Medium
Wicker, 2010 [102]	Cross-sectional study	2010 (Jan–May)	Frankfurt University Hospital, Germany	1504	Physicians 26%, nurses 35%, other HCW 23%, students 16%	None	Pertussis	22% in last 10 years	Medium
Wilson, 2019 [103]	Cross-sectional study	Influenza seasons 2015–2017	Southeast France	1539	74% hospital nurses, 26% community nurses	None	Seasonal influenza	Both seasons: 24% at least one season: 34%	Medium
Wilson, 2020 [104]	Cross-sectional study	2017–2018	Southeast France	1539	74% hospital nurses, 26% community nurses	None	Mandatory and recommended vaccines in France	96% BCG, 73% DTP (<10 years), 61% HBV, 58% pertussis, 64% measles, 39% VZV, 27% seasonal influenza (last year)	Medium
Zhang, 2011 [105] and 2012 [106]	Cross-sectional study	2010 (May-Oct.)	University Hospital London	522	Qualified nurses (79% working in hospital	None	Seasonal influenza	36%	Medium

* concerns vaccination uptake unless otherwise specified. ** Quality is assessed with the AXIS tool. A low score represents fulfillment of 1–9 out of 20 items, medium 10–14 and high 15–20 items (Exact scores are given in Appendix A). The following abbreviations are used (organized per column, in alphabetical order): Enrolment sites: USA = United States of America; UK = United Kingdom; NL = the Netherlands. Study populations: CD = Crohn’s Disease; GP = general practitioner; HCW = healthcare workers; HIV = human immunodefiency virus; HSCT = hematological stem cell transplantation; IBD = inflammatory bowel disease; ICP = immunocompromised patients; IS = immunosuppressive treatment; PHC = primary healthcare; RA = rheumatoid arthritis; SOT = solid organ transplantation; UC = colitis ulcerosa; VFR = travellers visiting friends and relatives. Theoretical frameworks: ASE = attitude, social influence and self-efficacy model; HBM = health belief model; KAP = knowledge, attitude, practice; HIM = the Health Intention Model; BIM = behavioral intention model; CME = Cognitive model of empowerment; RPA = risk perception attitude framework; Triandis = Triandis model of interpersonal behavior. Vaccinations: BCG = Bacillus Calmette-Guerin (vaccine for tuberculosis); DTP = diphtheria, tetanus, poliomyelitis; HAV = hepatitis A virus; HBV = hepatitis B virus; HZV = herpes zoster virus; JE = Japanese encephalitis; Men = meningococcal disease; menACWY = meningococcal serotype A, C, W and Y; MMR = measles, mumps, rubella; Pneu = pneumococcal disease; TBC = tuberculosis; Tdap = tetanus, diphtheria, acellular pertussis; VZV = varicella zoster virus, YF = yellow fever.

**Table 2 vaccines-08-00480-t002:**
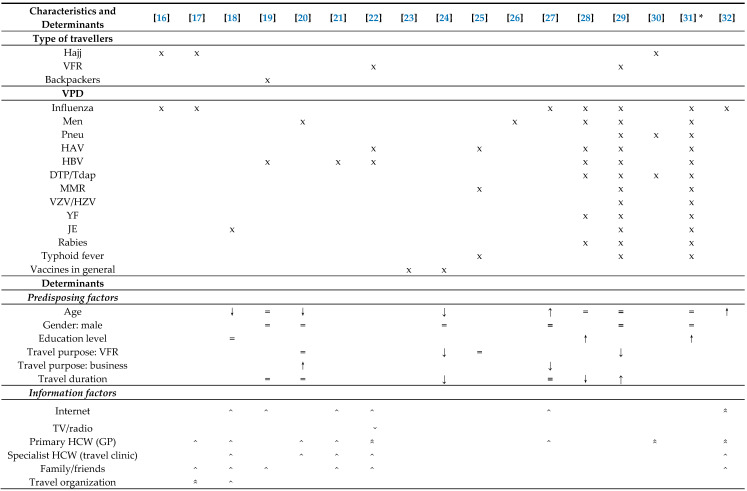
Overview of determinants of vaccination uptake in travellers.

The following symbols are used: x applicable; = no significant difference; **↑** significant positive association (tested by multivariate analysis); **↓** significant negative association (tested by multivariate analysis); ↑ significant positive association (tested by chi-square, univariate analysis or correlation coefficient); ↓ significant negative association (tested by chi-square, univariate analysis or correlation coefficient); 

 (double caret pointing upwards) significance was not tested, but determinant was positively linked to vaccination uptake in ≥50% of the population; 

 (double caret pointing downwards) significance was not tested, but determinant was negatively linked to vaccination uptake in ≥50% of the population; ⌃ (caret pointing upwards) significance was not tested, but determinant was positively linked to vaccination uptake in ≥10% of the population; ⌄ (caret pointing downwards) significance was not tested, but determinant was negatively linked to vaccination uptake in ≥10% of the population. * determinants were studied in relation to intention to be vaccinated instead of vaccination uptake. The following abbreviations are used (in alphabetical order): CD = Crohn’s Disease; DTP = diphtheria, tetanus, poliomyelitis; GP = general practitioner; HAV = hepatitis A virus; HBV = hepatitis B virus; HCW = healthcare workers; HIV = human immunodefiency virus; HSCT = hematological stem cell transplantation; HZV = herpes zoster virus; IBD = inflammatory bowel disease; IS = immunosuppressants; JE = Japanese encephalitis; Men = meningococcal disease; MMR = measles, mumps, rubella; Pneu = pneumococcal disease; Tdap = tetanus, diphtheria, acellular pertussis; SOT = solid organ transplantation; VFR = travellers visiting friends and relatives; VZV = varicella zoster virus; YF = yellow fever.

**Table 3 vaccines-08-00480-t003:**
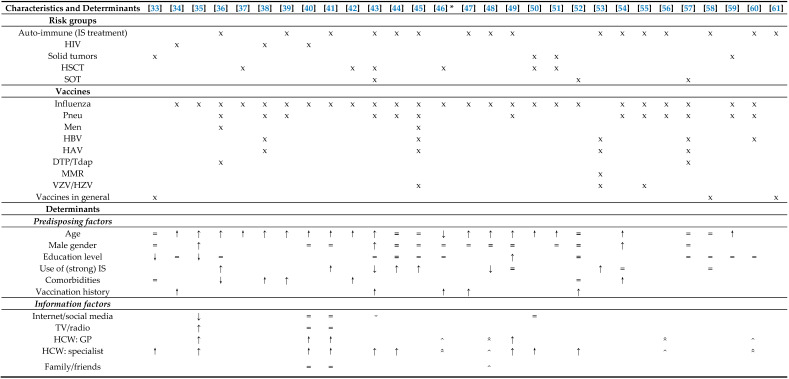
Overview of determinants of vaccination uptake in ICP.

The following symbols are used: x applicable; = no significant difference; **↑** significant positive association (tested by multivariate analysis); **↓** significant negative association (tested by multivariate analysis); ↑ significant positive association (tested by chi-square, univariate analysis or correlation coefficient); ↓ significant negative association (tested by chi-square, univariate analysis or correlation coefficient); 

 (double caret pointing upwards) significance was not tested, but determinant was positively linked to vaccination uptake in ≥50% of the population; 

 (double caret pointing downwards) significance was not tested, but determinant was negatively linked to vaccination uptake in ≥50% of the population; ⌃ (caret pointing upwards) significance was not tested, but determinant was positively linked to vaccination uptake in ≥10% of the population; ⌄ (caret pointing downwards) significance was not tested, but determinant was negatively linked to vaccination uptake in ≥10% of the population. The following abbreviations are used (in alphabetical order): CD = Crohn’s Disease; DTP = diphtheria, tetanus, poliomyelitis; GP = general practitioner; HAV = hepatitis A virus; HBV = hepatitis B virus; HCW = healthcare workers; HIV = human immunodefiency virus; HSCT = hematological stem cell transplantation; HZV = herpes zoster virus; IBD = inflammatory bowel disease; IS = immunosuppressants; JE = Japanese encephalitis; Men = meningococcal disease; MMR = measles, mumps, rubella; Pneu = pneumococcal disease; Tdap = tetanus, diphtheria, acellular pertussis; SOT = solid organ transplantation; VFR = travellers visiting friends and relatives. VZV = varicella zoster virus, YF = yellow fever.

**Table 4 vaccines-08-00480-t004:**
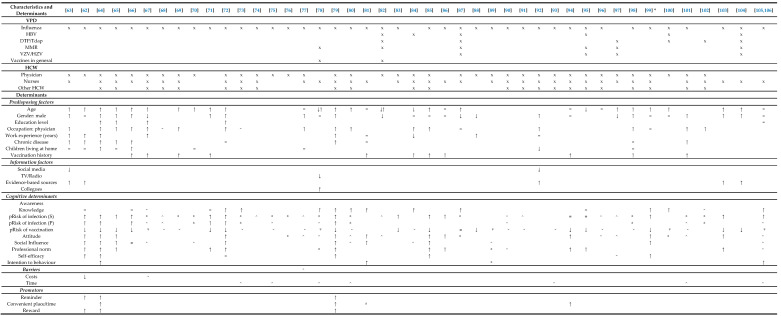
Overview of determinants of vaccination uptake in HCW.

* One scale (MoVac-flu scale) was used for following determinants: knowledge, attitude and self-efficacy. The following symbols are used: x applicable; = no significant difference; **↑** significant positive association (tested by multivariate analysis); **↓** significant negative association (tested by multivariate analysis); ↑ significant positive association (tested by chi-square, univariate analysis or correlation coefficient); ↓ significant negative association (tested by chi-square, univariate analysis or correlation coefficient); **↓**↑ significant association, for one vaccine positive, for the other negative; 

 (double caret pointing upwards) significance was not tested, but determinant was positively linked to vaccination uptake in ≥50% of the population; 

 (double caret pointing downwards) significance was not tested, but determinant was negatively linked to vaccination uptake in ≥50% of the population; ⌃ (caret pointing upwards) significance was not tested, but determinant was positively linked to vaccination uptake in ≥10% of the population; ⌄ (caret pointing downwards) significance was not tested, but determinant was negatively linked to vaccination uptake in ≥10% of the population. pRisk = perceived risk. pRisk of infection (S/P): S = self; P = patient. The following abbreviations are used (in alphabetical order): CD = Crohn’s Disease; DTP = diphtheria, tetanus, poliomyelitis; GP = general practitioner; HAV = hepatitis A virus; HBV = hepatitis B virus; HCW = healthcare workers; HIV = human immunodefiency virus; HSCT = hematological stem cell transplantation; HZV = herpes zoster virus; IBD = inflammatory bowel disease; IS = immunosuppressants; JE = Japanese encephalitis; Men = meningococcal disease; MMR = measles, mumps, rubella; Pneu = pneumococcal disease; Tdap = tetanus, diphtheria, acellular pertussis; SOT = solid organ transplantation; VFR = travellers visiting friends and relatives. VZV = varicella zoster virus, YF = yellow fever.

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
