# Peer review of "Determinants of Vaccination Uptake in Risk Populations: A Comprehensive Literature Review"

_vaccines, 2020, doi:10.3390/vaccines8030480_

Round 1

Reviewer 1 Report

Dear Authors,

the manuscript submitted treats a very interesting and current topic. However, some aspects should be addressed before to consider the paper suitable for potential publication. 

ADD THE DATA RELATING TO THE VACCINATION COVERAGE OF THE THREE ANALYZED GROUPS. 

Author Response

We thank reviewer #1 for the given feedback. The vaccination coverage data was added to Table 1. The second last column shows the vaccination coverage per study. As the data differed a lot per study and the study populations were very heterogeneous (even within the three analysed groups), we decided to not combine this data.

Reviewer 2 Report

This is a well conceived and executed review of the literature on specific risk group recommended for various vaccines.  I have a few suggestions for improvement.

In the introduction, the mention of the SARS-CoV-2 virus or vaccine seems to be gratuitous.  There is no other mention of it in the remainder of the paper.  If it is to be introduced, then the discussion should contain reference to how the findings from other vaccines would or could apply to a SARS-CoV-2 vaccine.  Otherwise, it does not seem to fit.

Also in the discussion, I would like to see more specific recommendations for improving vaccine uptake both for all risk groups combined and for each risk group specifically.  

Author Response

This is a well conceived and executed review of the literature on specific risk group recommended for various vaccines.  I have a few suggestions for improvement.

Response of the authors:

We thank reviewer #2 for the positive feedback and we will address the suggestions given one-by-one hereafter.

In the introduction, the mention of the SARS-CoV-2 virus or vaccine seems to be gratuitous.  There is no other mention of it in the remainder of the paper.  If it is to be introduced, then the discussion should contain reference to how the findings from other vaccines would or could apply to a SARS-CoV-2 vaccine.  Otherwise, it does not seem to fit.

Response of the authors:

We agree with the reviewer that there should be a place for SARS-CoV-2 vaccine in the discussion. We now added this to the implications section in line 498: “Moreover, this understanding could help with the implementation of new vaccines for certain risk groups, for instance when a novel SARS-CoV-2 vaccine will be recommended for HCW.”

Also in the discussion, I would like to see more specific recommendations for improving vaccine uptake both for all risk groups combined and for each risk group specifically. 

Response of the authors:

We extended the implications section with recommendations for the risk groups in specific and in general. We now state (in line 480-490): “Therefore, multifactorial interventions are needed that address most important cognitive determinants. As these include awareness and risk perceptions, reminders and incidence data could help. Reminders for travellers could be disseminated in general media before holidays, while for ICP patient associations and HCW could play a role. To improve risk perceptions for the infections, cases of vaccine-preventable diseases should be made public. To decrease risk perceptions of negative effects of vaccinations (e.g. adverse events) new studies should compare the number of influenza-like illness in vaccinated and non-vaccinated groups. Furthermore, social norms can be included by making the decisions of vaccination uptake public. For example, in HCW trials have been implemented to test the effects of providing a pin that vaccinated HCW may wear that is saying “deliberately vaccinated”, which could affect both colleagues and patients [91].”

Reviewer 3 Report

The paper by Doornekamp and colleagues is a systematic review of literature on the determinants for vacine uptake/refusal in three risky population.

Overall I have been pleased to read a comprehensive paper like this, which gives a broad overview on this very interesting field also of my research topics.

As a methodologist I have particularly appreciated the search startegies and the strings. They are really exaustive and do not need any other suggestion by me.

The methodological parte is a little bit scarce in details for the processes. Following PRISMA statement I suggest the authors to explicitly declare the process of selection and data extraction with a brief description of the researchers involved and the protocol adopted, for example, to resolve disagreement between the different readers.

The results section is clear, as well as it is Table 1, in which everything is described in a good way.

What I found to be a good point for implementation of your work is a quality assessment. I would like to suggest the authors to improve the paper by adding a quality assessment sectionand use the proper tools to evaluate the quality of the studies (i.e. STROBE checklist) and give also a qualitative analysis of the papers found.

Author Response

The paper by Doornekamp and colleagues is a systematic review of literature on the determinants for vacine uptake/refusal in three risky population.

Overall I have been pleased to read a comprehensive paper like this, which gives a broad overview on this very interesting field also of my research topics.

As a methodologist I have particularly appreciated the search startegies and the strings. They are really exaustive and do not need any other suggestion by me.

Response of the authors:

We thank reviewer #2 for the positive feedback and the underlining of the importance of our work. We appreciate his or her time spent on reviewing our manuscript.

The methodological parte is a little bit scarce in details for the processes. Following PRISMA statement I suggest the authors to explicitly declare the process of selection and data extraction with a brief description of the researchers involved and the protocol adopted, for example, to resolve disagreement between the different readers.

Response of the authors:

We now elaborated on the data selection and extraction processes using the PRISMA checklist. We extended the data selection part with: “In case any doubt or disagreement between the two researchers who performed the study selection (LD and LL) arose, the specific papers were discussed in a plenary session with all co-authors.”

To the data extraction paragraph we added (line 128): “Extracted data was collected in Microsoft Excel 2016 and the presence and impact of determinants were rated in separate sheets per study group (LD and LL). Random samples were taken to check the data extraction and disagreements were discussed plenary with all co-authors.”
Hopefully these additions meet the reviewers’ expectations.

The results section is clear, as well as it is Table 1, in which everything is described in a good way.

What I found to be a good point for implementation of your work is a quality assessment. I would like to suggest the authors to improve the paper by adding a quality assessment section and use the proper tools to evaluate the quality of the studies (i.e. STROBE checklist) and give also a qualitative analysis of the papers found.

Response of the authors:

As the STROBE checklist is a descriptive tool for the studies qualities, we have chosen to use the AXIS tool, which works with scores. This tool is specifically designed for cross-sectional studies, as those in our review, and only includes items relevant to this design. As it contains 20 questions that can be answered with yes and no, it gives a quick overview of the quality of the 91 studies included in this review. We now described this in the methods under data selection (line 131) saying: “Furthermore, the quality of studies was assessed using the AXIS tool, which is a screening tool specifically designed for cross-sectional studies, as those in our review, and includes items relevant to this design.” and in the results (line 166): “The results of the quality assessment are presented in Supplementary Table S3.” We added these scores to a separate supplementary file (S3) and a resume to Table 1 (last column).